# Why Methodology Is Important: Coffee as a Candidate Treatment for COVID-19?

**DOI:** 10.3390/jcm9113691

**Published:** 2020-11-17

**Authors:** Yaniss Belaroussi, Paul Roblot, Nathan Peiffer-Smadja, Thomas Delaye, Simone Mathoulin-Pelissier, Joffrey Lemeux, Gwenaël Le Moal, Eric Caumes, France Roblot, Alexandre Bleibtreu

**Affiliations:** 1INSERM, Bordeaux Population Health Research Center, ISPED, University of Bordeaux, 33000 Bordeaux, France; yaniss.belaroussi@chu-bordeaux.fr (Y.B.); paul.roblot@chu-bordeaux.fr (P.R.); simone.pelissier@u-bordeaux.fr (S.M.-P.); joffrey.lemeux@hotmail.com (J.L.); 2Department of Neurosurgery, Centre Hospitalier Universitaire de Bordeaux, 33000 Bordeaux, France; 3IAME, INSERM, Université de Paris, F-75006 Paris, France; n.peiffer-smadja@imperial.ac.uk; 4Service de Maladies Infectieuses et Tropicales, AP-HP Hôpital Bichat, F-75018 Paris, France; 5National Institute for Health Research, Health Protection Research Unit in Healthcare Associated Infections and Antimicrobial Resistance, Imperial College London, London SW7 2AZ, UK; 6Service de Maladies Infectieuses et Tropicales, CHU Poitiers, 86000 Poitiers, France; thomas.delaye@chu-poitiers.fr (T.D.); gwenael.le-moal@chu-poitiers.fr (G.L.M.); france.cazenave-roblot@chu-poitiers.fr (F.R.); 7INSERM CIC1401, Clinical and Epidemiological Research Unit, Institut Bergonié, 33000 Bordeaux, France; 8INSERM U1070, University of Poitiers, 86000 Poitiers, France; 9Assistance Publique—Hôpitaux de Paris, Hôpitaux Universitaires Pitié-Salpêtrière Charles Foix, Infectious Diseases Department, Pitié-Salpêtrière Hospital, 47–83 Boulevard de l’hôpital, 75013 Paris, France; eric.caumes@aphp.fr; 10INSERM, Institut Pierre Louis d’Épidémiologie et de Santé Publique, Sorbonne University, 75013 Paris, France; 11COVID SMIT PSL STUDY GROUP Infectious Diseases Department, Pitié-Salpêtrière Hospital, 47-83 Boulevard de l’hôpital, 75013 Paris, France

**Keywords:** coronavirus disease 2019 (COVID-19), current therapies, symptomatic treatment, trimethylxanthine, methodology

## Abstract

Background: During this pandemic situation, some studies have led to hasty conclusions about Corona Virus Disease-19 (COVID-19) treatment, due to a lack of methodology. This pedagogic study aimed to highlight potential biases in research on COVID-19 treatment. Methods: We evaluate the effect of coffee’s active part, 1,3,7-trimethylxanthine (TMX) on patients with COVID-19. A cohort of 93 patients, with a diagnosis of COVID-19 is analyzed. Results: TMX group and control group included, respectively, 26 and 67 patients. In the TMX group, patients had a median length of stay in hospital of 5.5 days shorter than in the control group (9.5 vs. 15 days, *p* < 0.05). Patients in the control group were more severe than patients in the TMX group with a significantly higher National Early Warning Score 2 (NEWS-2 score) (8 vs. 6, *p* = 0.002). Conclusions: Multiple biases prevents us from concluding to an effect of coffee on COVID-19. Despite an important social pressure during this crisis, methodology and conscientiousness are the best way to avoid hasty conclusions that can be deleterious for patients. Identifier: NCT04395742.

## 1. Introduction

A new pandemic respiratory infection caused by the coronavirus disease 2019 (COVID-19), the first cases of which were reported in Wuhan, China [1], killing approximately 0.5% of confirmed case mostly by acute respiratory failure [2]. On 15 August 2020, the French government reported that 30,406 patients died from COVID-19 on the national territory [3]. During the epidemic’s first months, several treatments were rapidly recommended, mainly as symptomatic treatments. Numerous anti-infectious treatments were rapidly under investigation [4], including remdesivir [4], lopinavir/ritonavir [5,6], or hydroxychloroquine [7]. Angiotensin-converting enzyme inhibitors during COVID-19 infection were also discussed [8,9].

Coffee contains 1,3,7-trimethylxanthine (TMX), which is thus the most commonly used psychoactive drug in the world. Recent studies suggested an efficacy of 1,3,7-trimethylxanthine (TMX) on the tolerance to central hypovolemia [9] and the prevention of hypotension-related syncope [10]. In a study with 13 patients, the exposure to lower body negative pressure led to a decrease of blood pressure that was significantly inferior in participants who took caffeinated coffee. Syncopes are typically caused by a decrease in blood flow, generally from low blood pressure. Consequently, improved blood pressure control allows a better perfusion of the brain and a reduced risk of syncope.

Due to the interest in drug repurposing as an avenue for research in COVID-19 crisis [11] and the large number of clinical studies aiming to evaluate treatments using inappropriate design and methods, we conducted a pedagogic comparative study aiming to evaluate how the lack of proper methodology can suggest the efficacy of coffee (1,3,7-trimethylxanthine) for the treatment of patients with COVID-19.

## 2. Population and Methods

### 2.1. Study

Patients from two centers (Centre Hospitalier Universitaire La Pitié-Salpêtrière, Paris, France and Centre Hospitalier Universitaire de Poitiers, Poitiers, France) were included during April 2020. They were assessed for eligibility at the hospital admission in the same center. The Research Ethics Committee of the Sorbonne University approved this study (CER-2020-30), personal data has been stored after authorization by the Commission Nationale de l’Informatique et des Libertés (CNIL), French data protection authority (declaration numbers 2085881 and 2211250v0) and is registered with ID-RCB number 2020-A01410-39 and ClinicalTrials.gov ID: NCT04395742. The study was designed as a pedagogic study to show that inappropriate methodology and biases can wrongly conclude to the efficacy of a candidate drug.

### 2.2. Patients

All patients with a confirmed COVID-19 infection were assessed for the following inclusion criteria: age > 18 and no TMX contraindications (defined as allergy and previous reported adverse events). Before inclusion, all patients had to give their consent. These patients were all proposed an oral TMX dose every morning, during breakfast. TMX was delivered in the morning coffee at an estimated dose of 65 mg of caffeine per day. The absorbed dose may vary between participants due to incomplete coffee consumption. The control group was based on patients who refused the TMX dose, who could not orally take the treatment, and the patients who were administered corticosteroids due to the interactions between TMX and steroids [12]. This group was defined as a control group. All patients received symptomatic treatment.

COVID-19 infection diagnosis was confirmed after a thoracic CT-scan showing typical patterns as multifocal consolidative opacities, septal thickening, or crazy paving [13] and/or a nasopharyngeal swab with a PCR array. All data were assessed by infectious disease specialists. The severity of the symptoms were defined with National Early Warning Score 2 (NEWS-2) [14].

### 2.3. Assessments

The primary outcome measure was the patient clinical status at day 6 after hospital admission, defined as alive or dead. The secondary judgments criteria were duration of hospital stay, severity evaluated by NEWS-2 score, secondary effects independent from the disease, and use of antibiotics during the hospital stay.

### 2.4. Statistical Analysis

Qualitative variables were described by counts and percentages, quantitative variables were described by either their mean and standard deviation (SD) when their distributions were normal or median and interquartile range (IQR). Patients from TMX-group and patients from control group were compared using the Chi-square test or the Fisher’s exact test for qualitative variables and using the *t*-test or the Wilcoxon test for quantitative variables. A threshold of α = 0.05 was considered for significance. 

Analyses were performed with RStudio software version 3.5.1 R Foundation for Statistical Computing, Vienna, Austria. ISBN 3-900051-07-0, URL http://www.R-project.org.

## 3. Results

### 3.1. Population

Within a period of two weeks, 93 patients were consecutively hospitalized in both centers with COVID-19 confirmed by PCR and/or a thoracic CT-scan. The mean age was 67.3 years (SD = 17.1) and the mean body mass index (BMI) was 27.8 kg/m^2^ (SD = 6.8). About half of the patients presented hypertension. Diabetes was found in a quarter of the population. Pulmonary disease was less frequent (14% of the patients). Most of the patients had never smoked. The TMX group and the control group included, respectively, 26 and 67 patients. No significant difference in gender, BMI, associated diseases, smoking status, or usual antihypertensive treatment between TMX and control groups were found (Table 1). The mean age in the TMX group seemed to be a little higher but significance was not reached (72.7 vs. 65.3 years, *p* = 0.06). Patients in the control group were more severe as shown by the NEWS-2 score which was significantly higher in the control group (8 vs. 6, *p* = 0.002).

### 3.2. Outcomes

Patients receiving the TMX treatment had a significantly shorter hospital stay (9.5 vs. 15 days, *p* = 0.027) (Table 2). Use of antibiotics was significantly lower in the TMX group as compared to the control group (46.2% vs. 73.1%, *p* = 0.014). CT-scans showed that the extension of the lesions was less important in the TMX group than in the control group (17.5% vs. 37.5%, *p* < 0.001). The multivariate regression demonstrated that patients from the TMX group had an odd to be hospitalized less than seven days, about five times higher than patients from control group, adjusted on gender, obesity status, and age (Table 3).

## 4. Discussion

Patients receiving the TMX treatment had a significantly shorter hospital stay (9.5 vs. 15 days, *p* < 0.05). Moreover, TMX decreased the use of antibiotics and the extension of pulmonary lesions in the CT-scans. These results lead to the conclusion that the 1,3,7-Trimethylxanthine (TMX), the active molecule of the patients’ morning coffee beverage, is a potential treatment for COVID-19 infection.

In fact, we cannot conclude to a direct effect of TMX on COVID-19 because of a huge lack of methodological considerations [15]. The aim of this study was not to prove the efficacy of coffee but to reveal how anyone can prove anything with a non-rigorous study. For instance, the multiple bias, the selection criteria, the clinical relevance of the endpoints, and the methodological structure of our study weakened the external and internal validity.

First, the lack of accurate selection and exclusion criteria led to the production of two incomparable groups. Even if those two groups seem to be similar according to demographic characteristics (Table 1), the TMX group was clearly advantaged by its construction. Patients who could not orally take TMX were considered only in the control group and so were all the patients who needed mechanical ventilation. The same trick was used to move patients treated with corticosteroids from TMX group to the control group. This has been observed in several published articles evaluating candidate drugs for the treatment of patients with COVID-19 [16]. Confounding factors must not be overlooked as they are strongly associated with both exposure and outcomes.

Second, a constant lack in observational studies is the indication bias [17]. Why does a patient receive this treatment instead of another? A statistical pathway, such as a propensity score, can be discussed even if it is not a magical solution and has many limitations, such as it does not take into consideration unnotified factors [17]. Indication bias can be avoided with the use of randomization.

Third, the primary outcome defined in the methods as the mortality at day six after hospital admission is not reported in the results. Indeed, mortality was not different between the control and the treatment group (6% vs. 7.7%, *p* = 0.671, Table 2) but the results only described secondary outcomes. Focusing on secondary outcomes is frequent in controlled trials and is a type of spin defined as “exaggerating the clinical significance of a particular treatment without the statistics to back it up”. Spin has been widely described in biomedical literature and is a questionable research practice [18,19,20]. Moreover, there was no estimation of the number needed to treat, the power, or the expected effect of the treatment described in the methods. Most of the recommendations of the STROBE guidelines for reporting observational studies were not followed in the current study [21].

Fourth, temporality is one of the keys to determining a causality between exposure (or treatment) and outcome. To conclude that TMX decreased the extension of lung lesions is inappropriate and does not fit with Bradford Hill criteria for causality [22].

Finally, the multivariable analysis suggested that TMX is associated with a higher odd to be hospitalized for less than seven days. Adjustment with actual known prognostic factors (age, obesity, and gender) [18,23] does not mean causality. For instance, selection of the covariates for an appropriate analysis of causal relationship can be done with a directed graph [19]. This approach can eventually reduce the degree of the bias in the estimated model [20]. In this study, including initial clinical status would change all the results. We should stay cautious with the use of statistical models and the results as the knowledge in this pandemic evolves day after day, particularly with small sample sizes. The strategy of analysis depends on the conception of the study and unfortunately, without a relevant methodology, prognosis and causality can be confounded [21].

Well-conducted studies are important to make reliable conclusions. Limitations of the studies must be considered from their conception to their conclusion. 

Controlled randomized clinical trials are considered as the gold standard to evaluate treatment effects [24]. To control biases, randomized trials provide a fair allocation of both known and unknown risk factors to each group. Both blinded follow-up and outcome evaluation sustain comparability between the groups [25]. As a result, in a randomized trial with intention-to-treat analyses, conclusions on outcomes can be assigned to the causal effect of the treatment [26]. Well-conducted observational studies can be useful to describe a scientific idea but must be considered with cautiousness as they can easily be biased [27].

Unfortunately, the COVID-19 pandemic crisis situation conducted to an increasing use of methodological-lacking observational studies or poorly-designed clinical trials to evaluate the effects of treatment [28,29]. Emergency statements have conducted to a significant belief bias as “people require less evidence to endorse a syllogism as valid when it has a believable conclusion” [30]. However, no treatment of COVID-19 has been found and so grows the desperation to find a “magic pill” [31,32,33,34,35,36,37]. 

The fundamental issue is to identify a factual clinical benefit in the general population. 

We cannot conclude of any association between coffee or TMX and COVID-19. We hope that nobody will use caffeine without medical surveillance and more generally, nobody will use caffeine against coronavirus. Overconsumption of caffeine, as with the overconsumption of any drug or molecule, could result in adverse events [38].

This study is not about a potential therapy in COVID-19 infection but a pedagogic study with real data from real patients treated for COVID-19 in two centers. We voluntarily conducted and analyzed this study to highlight that methodological-lacking studies can lead to overinterpreted data and give the wrong conclusions. We hope that an educational message is best conveyed through humor [39].

## 5. Conclusions

How can the most commonly used psychoactive drug in the world, TMX, the active molecule of coffee, treat COVID-19? With data from real patients, we tried to demonstrate that the lack of methodology and shortcuts can lead to inappropriate conclusions. We opine that carefulness, criticism, and collegiality are required to avoid pitfalls due to the multiplicity of articles on COVID-19. Despite an important social pressure during this crisis, methodology and conscientiousness are the best way to avoid bias and hasty conclusions. 

## Figures and Tables

**Table 1 jcm-09-03691-t001:** Characteristics of the 93 patients at baseline.

	Overall (*n* = 93)	TMX Group (*n* = 26)	Control Group (*n* = 67)	*p*-Value
Gender (*n*, %)				0.788
Male	48 (51.6)	14 (53.8)	33 (49.3)	
Female	45 (48.4)	12 (46.2)	34 (50.7)	
Age (mean ± SD)	67.3 ± 17.1	72.7 ± 16.4	65.3 ± 17.1	0.060
BMI (mean ± SD)	27.8 ± 6.8	25.7 ± 5.2	28.6 ± 7.1	0.096
Associated disease (*n*, %)				
Hypertension	49 (52.7)	14 (53.8)	35 (52.2)	0.889
Diabetes	23 (24.7)	4 (17.4)	19 (28.4)	0.193
Pulmonary disease (asthma, COPD)	13 (14.0)	5 (19.2)	8 (11.9)	0.369
Hepatic or renal chronic failure	11 (11.8)	4 (15.4)	7 (10.4)	0.761
Smoking status (*n*, %)				0.414
Never smoked	68 (73.1)	18 (59.2)	50 (74.6)	
Former smoker	18 (19.4)	7 (26.9)	11 (16.4)	
Current smoker	7 (7.5)	1 (3.8)	6 (9.0)	
Usual antihypertensive treatment (*n*, %)				
Beta-blocker	23 (24.7)	6 (23.1)	17 (25.4)	0.788
Angiotensin-Converting Enzyme inhibitors	17 (18.3)	7 (26.9)	10 (14.9)	0.179
Angiotensin-Receptor blockers	14 (15.1)	3 (11.5)	11 (16.4)	0.970
Initial NEWS score (median, IQR)	7 (4.5;9)	6 (4;7)	8 (5.3;10)	0.002

IQR: Interquartile range.

**Table 2 jcm-09-03691-t002:** Outcomes of the 93 patients.

	TMX Group (*n* = 26)	Control Group (*n* = 67)	*p*-Value
Use of antibiotics (*n*, %)	12 (46.2)	49 (73.1)	0.014
Percentage of lung lesion in CT-scan (median, IQR)	17.5 (10;35)	37.5 (17.5;60)	<0.001
Length of stay in hospital in days (median, IQR)	9.5 (6.3;18.3)	15 (9;24)	0.027
Deaths (*n*, %)	2 (7.7)	4 (6.0)	0.671

IQR: Interquartile range.

**Table 3 jcm-09-03691-t003:** Evaluation of the odds of length of stay in hospital < 7 days, adjusted on demographic characteristics.

	Crude OR (95% IC)	Adjusted OR (95% IC)	*p*-Value
MTX group vs. control group (ref)	3.91 (0.93; 16.33)	4.87 (1.02; 23.29)	0.044
Gender: Male vs. Female (ref)	1.88 (0.43; 8.17)	2.20 (0.46; 10.62)	0.311
Obese vs. non-obese (ref)	0.57 (0.11; 2.98)	0.71 (0.13; 3.99)	0.691
Age (years)	0.99 (0.95; 1.03)	0.97 (0.93;1.02)	0.267

OR: Odds ratio. Ref: Reference.

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
