# Peer review of "Why Methodology Is Important: Coffee as a Candidate Treatment for COVID-19?"

_jcm, 2020, doi:10.3390/jcm9113691_

Round 1
Reviewer 1 Report
I applaud the authors for finding a creative and highly amusing way to illustrate important methodological problems that have been pervasive in the COVID literature and concerning to many researchers. I look forward to being able to use this article when teaching graduate students.
I only have two recommended revisions:
1- I am a bit concerned that the Introduction, Methods, and Results do not have strong statements indicating that this is a pedagogic study. Perhaps this is something that can be resolved in the running head? Based on prior incidents with the BMJ Christmas articles, I think a stronger warning somewhere in the Intro is warranted. The discussion makes it quite clear, but my concern is that it's possible for someone who only reads the first few sections to take the paper seriously.
2- Line 119-121: These points could use a slightly expanded explanation. They get lost in the larger context of the discussion, but I think clarifying further the poor precision/small sample size and the choice of the primary outcome would be useful.
Author Response
In italics, the reviewers’ comments, in bold the authors’ response and modifications in the manuscript are in bold and underlined.
Reviewer 1:
I applaud the authors for finding a creative and highly amusing way to illustrate important methodological problems that have been pervasive in the COVID literature and concerning to many researchers. I look forward to being able to use this article when teaching graduate students.
Authors’ reponse:
We thank the reviewer for these kind comments and indeed, when we designed this study and article, we thought that it could be useful for teaching students.
I only have two recommended revisions:
1- I am a bit concerned that the Introduction, Methods, and Results do not have strong statements indicating that this is a pedagogic study. Perhaps this is something that can be resolved in the running head? Based on prior incidents with the BMJ Christmas articles, I think a stronger warning somewhere in the Intro is warranted. The discussion makes it quite clear, but my concern is that it's possible for someone who only reads the first few sections to take the paper seriously.
Authors’ reponse:
We fully agree with the reviewer and we do not want people to believe that coffee is an effective treatment for COVID-19. We have seen so much misinformation during this crisis (the so-called “infodemic”) that we have to take extra care and be very clear on the nature of this study.
We have introduced the following statements:
In the abstract:
Patients in the control group were more severe than patients in the TMX group with a significantly higher NEWS-2 score (8 vs 6, p=0.002).
In the introduction:
Due to the interest in drug repurposing as an avenue for research in COVID-19 crisis[11] and the large number of clinical studies aiming to evaluate treatments using inappropriate design and methods, we conducted a pedagogic comparative study aiming to evaluate how the lack of proper methodology can suggest the efficacy of coffee (1,3,7-trimethylxanthine) for the treatment of patients with COVID-19.
In the methods:
The study was designed as a pedagogic study to show that inappropriate methodology and biases can wrongly conclude to the efficacy of a candidate drug.
In the results:
Patients in the control group were more severe as shown by the NEWS-2 score which was significantly higher in the control group (8 vs 6, p=0.002).
2- Line 119-121: These points could use a slightly expanded explanation. They get lost in the larger context of the discussion, but I think clarifying further the poor precision/small sample size and the choice of the primary outcome would be useful.
Authors’ reponse:
We agree with the reviewer and these points are major to explain the importance of appropriate methodology. We have expanded the discussion following your advice:
Third, the primary outcome defined in the methods as the mortality at day 6 after hospital admission is not reported in the results. Indeed, mortality was not different between the control and the treatment group (6% vs 7.7%, p = 0.671, Table 2) but the results only described secondary outcomes. Focusing on secondary outcomes is frequent in controlled trials and is a type of spin defined as “exaggerating the clinical significance of a particular treatment without the statistics to back it up”. Spin has been widely described in biomedical literature and is a questionable research practice.(20–22) Moreover, there was no estimation of the number needed to treat, the power or the expected effect of the treatment described in the methods. Most of the recommendations of the STROBE guidelines for reporting observational studies were not followed in the current study.(23)
Reviewer 2 Report
The manuscript by Belaroussi et al describes the importance of methodological design when interpreting the efficacy of a drug, in this case caffeine in coffee, upon COVID-19 outcome measures. This study has been poorly designed on purpose in order to highlight the pitfalls of weak study design in the appropriate evaluation of a repurposed drug in COVID 19 treatment. This study is novel and very relevant. The authors are addressing an important issue in this paper and I enjoyed reading it. I have one major concern which is easy to fix, and a few minor concerns that can also easily be addressed.
Major:
The ethical approval information presented in lines 153 -157 needs to be moved to the method section (first paragraph).
Minor:
Line 45: Please consider explaining the relevance of an improved BP control during central hypovolemia and prevention of hypotension related syncope to COVID-19.
Line 57: Detail about the oral TMX dose is scant (perhaps deliberately so). That said, please provide more information here such as the method of TMX delivery (presumably coffee) and the estimated dose of caffeine given in that coffee etc.
I understand that the aim of this manuscript is to convey to the reader that experimental design is of paramount importance when examining the efficacy of any repurposed medication/drug upon COVID-19 outcome measures. In reading the discussion, this needs to come across in the first paragraph rather than the second paragraph. Even though the authors clearly point out that there is no association between coffee and COVID 19 (Line 149) it appears deeper into the body of the discussion. I have a minor concern that some readers may still misinterpret the point of this manuscript and if they were to only read the first paragraph they might, through no fault of the authors, take away information that the authors clearly do not intend.
Line 165: suggest substituting the word "inadequate" with "inappropriate".
Author Response
Reviewer 2:
The manuscript by Belaroussi et al describes the importance of methodological design when interpreting the efficacy of a drug, in this case caffeine in coffee, upon COVID-19 outcome measures. This study has been poorly designed on purpose in order to highlight the pitfalls of weak study design in the appropriate evaluation of a repurposed drug in COVID 19 treatment. This study is novel and very relevant. The authors are addressing an important issue in this paper and I enjoyed reading it. I have one major concern which is easy to fix, and a few minor concerns that can also easily be addressed.
Authors’ reponse:
We thank the reviewer for this kind assessment of our study and his/her suggestions that are valuable to improve the manuscript.
Major:
The ethical approval information presented in lines 153 -157 needs to be moved to the method section (first paragraph).
Authors’ reponse:
Thank you. This was a mistake and we have moved the ethical approval information in the methods section.
Minor:
Line 45: Please consider explaining the relevance of an improved BP control during central hypovolemia and prevention of hypotension related syncope to COVID-19.
Authors’ reponse:
This is indeed an interesting point and we have expanded the discussion on this:
Recent studies suggested an efficacy of 1,3,7-trimethylxanthine (TMX) on the tolerance to central hypovolemia[9] and the prevention of hypotension-related syncope[10]. In a study with 13 patients, the exposure to lower body negative pressure led to a decrease of blood pressure that was significantly inferior in participants who took caffeinated coffee. Syncopes are typically caused by a decrease in blood flow to generally from low blood pressure. Consequently, An improved blood pressure control allows a better perfusion of the brain and a reduced risk of syncope,
Line 57: Detail about the oral TMX dose is scant (perhaps deliberately so). That said, please provide more information here such as the method of TMX delivery (presumably coffee) and the estimated dose of caffeine given in that coffee etc.
Authors’ reponse:
Indeed, we should have been more precise and we have included the estimated dose and mode of TMX delivery.
TMX was delivered in the morning coffee at an estimated dose of 65 mg of caffeine per day. The absorbed dose may vary between participants due to incomplete coffee consumption.
I understand that the aim of this manuscript is to convey to the reader that experimental design is of paramount importance when examining the efficacy of any repurposed medication/drug upon COVID-19 outcome measures. In reading the discussion, this needs to come across in the first paragraph rather than the second paragraph. Even though the authors clearly point out that there is no association between coffee and COVID 19 (Line 149) it appears deeper into the body of the discussion. I have a minor concern that some readers may still misinterpret the point of this manuscript and if they were to only read the first paragraph they might, through no fault of the authors, take away information that the authors clearly do not intend.
Authors’ reponse:
We fully agree with the reviewer and that was a point also highlighted by the other reviewer. We do not want people to believe that coffee is an effective treatment for COVID-19. We have seen so much misinformation during this crisis (the so-called “infodemic”) that we have to take extra care and be very clear on the nature of this study.
We have introduced the following statements in the manuscript:
In the abstract:
Patients in the control group were more severe than patients in the TMX group with a significantly higher NEWS-2 score (8 vs 6, p=0.002).
In the introduction:
Due to the interest in drug repurposing as an avenue for research in COVID-19 crisis[11] and the large number of clinical studies aiming to evaluate treatments using inappropriate design and methods, we conducted a pedagogic comparative study aiming to evaluate how the lack of proper methodology can suggest the efficacy of coffee (1,3,7-trimethylxanthine) for the treatment of patients with COVID-19.
In the methods:
The study was designed as a pedagogic study to show that inappropriate methodology and biases can wrongly conclude to the efficacy of a candidate drug.
In the results:
Patients in the control group were more severe as shown by the NEWS-2 score which was significantly higher in the control group (8 vs 6, p=0.002).
Line 165: suggest substituting the word "inadequate" with "inappropriate".
Authors’ reponse:
Thank you. We have modified according to the reviewer’s comment.